# Influence of Sensation Seeking and Life Satisfaction Expectancy on Stock Addiction Tendency: Moderating Effect of Distress Tolerance

**DOI:** 10.3390/bs13050378

**Published:** 2023-05-04

**Authors:** Myounghwan Son, Goo-Churl Jeong

**Affiliations:** 1Department of Counseling Psychology, Graduate School, Sahmyook University, Seoul 01795, Republic of Korea; sonwooju@gmail.com; 2Department of Counseling Psychology, College of Health and Welfare, Sahmyook University, Seoul 01795, Republic of Korea

**Keywords:** stock addiction, sensation seeking, life satisfaction expectancy, distress tolerance

## Abstract

Due to the COVID-19 pandemic, a very low interest rate policy was economically applied in Korea, and various investment activities through loans were activated. Real estate and stock prices rose rapidly, and many people became involved in stock investments because of economic instability. However, hastily started investment behavior resulted in economic loss and addictive behavior in stocks. The phenomenon of using stock investment to satisfy individual sensation seeking or addictive dependence on stocks due to lowered life satisfaction expectancy can become a serious social problem. However, the improvement of distress tolerance and the ability to endure pain despite frequent stock price fluctuations or lowered life satisfaction expectancy would be good alternatives to prevent stock addiction tendency. Therefore, the purpose of this study is to test the moderating effect of distress tolerance on the effect of adults’ sensation seeking and life satisfaction expectancy in stock addiction tendencies. The participants were 272 adults with stock investment experience. As a result, distress tolerance significantly moderated the positive effect of sensation seeking on stock addiction tendency. In addition, life satisfaction expectancy did not significantly increase in the group with high distress tolerance even if life satisfaction expectancy was lowered. These results suggest that stock addiction can be prevented by enhancing distress tolerance.

## 1. Introduction

In January 2021, in the United States, there was an incident in which the stock price of GameStop, a video game store, soared by 994.4% from USD 17.69 in early January to USD 193.6 at the end of January [1]. Not only American investors but also investors around the world focused on this case, and as it became known as a big issue in Korea, it soon led to direct investment activities. Since then, GameStop’s stock price has gone up and down significantly, and investors who have invested in the stock have cheered and felt a sense of pleasure, not only in terms of financial gains but also in observing the stock price, which has risen significantly. Considering the unpredictability of stock price fluctuations, this was not a healthy stock investment but rather a gamble swayed by emotion, luck, and desire [2]. According to a report by the Korea Financial Investment Association (2022), as of 30 September 2022, the number of stock accounts with deposited assets of approximately USD 85 or more and at least one transaction in the last six months was 63,464,663 [3]. This means that, when calculated based on the resident population of Korea as of September 2022 (51,466,658), the number of stock accounts held per citizen was 1.23 [4]. This shows that Korean stock investment activities are very active, and enthusiasm for stock investment is high. However, on the dark side of stock investment, “don’t ask” investment that seeks short-term returns, even at risk, and excessive stock investment behavior that attracts and invests all kinds of funds is also increasing [5]. Excessive investment without sufficient knowledge and assests puts investors at greater economic risk, causing depression or leading to tragedies, such as suicide, which is a socially worrisome problem [6].

Moreover, as interest rates sharply lowered after the COVID-19 pandemic and led to a craze for stock investment using loans, the number of people in their 20s and 30s suffering from addiction-like problems related to stock investment has increased significantly. According to a report by the Korea Institute for Prevention and Healing of Gambling Problems, it was found that the number of counseling requests related to stock addiction increased significantly from 3540 in 2020 to 5524 in 2021 [7].

Unlike gambling, which relies on luck, stocks are a concept of investing in a company. However, there is contingency because stock prices are affected by economic prospects, interest rates, or international situations, and there is pleasure related to the feeling of superiority when one’s prediction is correct resulting in a large financial reward. In addition, stocks have spatiality and capitalism, and trading is carried out through capital in the stock market. Because of the contingency, playfulness, spatiality, and capitalism of stocks, some scholars have argued that stock investment has characteristics similar to gambling [8]. Concepts such as excessive immersion in stock investment or psychological pleasure from short-term gains show similar characteristics to those of some gambling addictions. Both pathological gamblers and excessive stock investors have a financial motive for easy gains or to make up for lost money, an arousal motive for thrills and pleasures, and an avoidance motive for avoiding real-world problems that cause feelings of pressure or depression [8,9].

Zuckerman defined sensation seeking as a desire to pursue a variety of stimulating sensations or experiences, even if one is willing to take social, legal, physical, and financial risks [10]. According to previous studies, it has been reported that people with a high sensation seeking tendency, such as seeking thrills or pleasures, are disposed to addictive behaviors such as problematic drinking, pathological gambling, and excessive immersion on the internet [11,12,13,14,15]. However, no study has verified whether sensation seeking tendencies have a similar effect on stock addiction tendencies showing addictive behaviors; therefore, research on this is necessary.

On the other hand, the media has pointed out that the anxiety of “it has to be now” is an important factor in the recent overheating of stock investment [7]. When real estate prices rise rapidly and positive life satisfaction expectancy disappears, a speculative stock investment pattern to create large profits at once may appear. Real problems, such as deepening economic disparity, fixed hierarchical order, and unemployment in our society, are adding to this anxiety. To reduce this anxiety, we try to take measures such as the rapid accumulation of wealth [16]. Therefore, if one’s positive life satisfaction expectancy for the future is lowered, it can lead to inappropriate forms of stock investment in pursuit of short-term returns. Although no study has directly verified the effect of life satisfaction expectancy on stock addiction tendency, many previous studies have reported that it has a significant negative correlation with internet gaming and smartphone addiction [17,18]. Therefore, it is expected that it is closely related to stock addiction.

However, even if an individual’s sensation seeking is high and life satisfaction expectancy is low, an increase in stock addiction tendency can be prevented if the distress tolerance to endure negative situations or emotions is high. Distress tolerance refers to the ability to endure painful internal experiences such as physical discomfort or negative emotions [19,20]. Previous studies have reported that the lower the distress tolerance, the higher the SNS addiction tendency and the possibility of maintaining drinking problems [21,22]. People with sufficient economic knowledge and well-equipped stock investment skills can be resilient to changes in stock prices; however, unprepared investors can suffer extreme psychological distress. People who do not have the strength to endure psychological pain may have an increased tendency toward stock addiction.

Our society is aware of danger signals about some gambling and addictive aspects of stock investment, but there is no clear social and academic definition of stock addiction. Although stock investment can be viewed from the perspective of addiction through research on the gambling nature of stock investment and the development of a stock addiction scale [8,23,24], there are very few studies on stock addiction tendency. Therefore, care must be taken not to diagnose stock addiction categorically.

This study aims to analyze the effect of sensation seeking, which corresponds to individual psychological needs, and positive life satisfaction expectancy on stock addiction tendency, and to verify the moderating effect of distress tolerance. Through this, we intend to provide the basic data necessary to prepare measures to prevent stock addiction tendencies, which have recently emerged as a social problem.

The hypotheses of this study are as follows.

**Hypothesis** **1.**
*Sensation seeking, life satisfaction expectancy, and distress tolerance will significantly explain stock addiction tendency.*


**Hypothesis** **2.**
*There will be a moderating effect of distress tolerance in the effect of sensation seeking on stock addiction tendency.*


**Hypothesis** **3.**
*There will be a moderating effect of distress tolerance in the effect of life satisfaction expectancy on stock addiction tendency.*


## 2. Materials and Methods

### 2.1. Participants

Participants in this study were adults aged 19 years or older with experience in stock investment. Those who found it technically difficult to participate in the online survey or older than 65 were excluded from the study. We collected data from 250 people through an online survey, removed 23 data that answered insincerely about age, etc., and analyzed the final 227 people’s data.

The study participants included 37.4% males and 62.6% females. The age distribution ranged from 19 to 65 years, with an average age of 34.6 years (±8.3). As for the occupations of the research participants, 32.2% were professional/technical workers, followed by office workers (26.9%), sales/services workers (21.1%), students (13.7%), and the unemployed (6.2%). Among the study participants, 52.4% were married, 45.8% were unmarried, and 1.8% were other.

Looking at the characteristics related to the stock investment of the research participants, 83 (36.6%) of the respondents answered that their investment type was short-term, which includes buying and selling several times in a short period and aims for market profit. A total of 144 people (63.4%) responded that their long-term investment type holds investment assets for a long time. As a result of asking about the profit and loss of stock investment over the past year, 51 (22.5%) respondents said they had a loss, 83 (36.6%) reported being average, and 93 (41.0%) said they had made a profit. As a result of surveying satisfaction with the current stock investment, 57 people (25.1%) were dissatisfied, 91 (40.1%) were normal, and 79 (34.8%) were satisfied.

### 2.2. Procedure

This study was approved by the Institutional Review Board of Sahmyook University (IRB No. 2-1040781-A-N-012021066HR). The survey was conducted between 28 June and 15 July 2021 through an online survey promoted through stock-related community sites and social media after creating an online questionnaire using Google Forms. It was designed so that it started only when voluntary participation in the study was agreed upon. In addition, the storage method, period, and destruction method of the collected data were announced, and it was noted that participants could withdraw at any time if they experienced psychological discomfort or wanted to stop participating in the study. We sent participants an online coupon worth USD 1.50 as a reward for their time participating in the survey. 

### 2.3. Instrument

#### 2.3.1. Sensation Seeking

The Korean version [25] of the Brief Sensation Seeking Scale (BSSS) by Hoyle, Stephenson, Palmgreen, Lorch, and Donohew (2002) [26] was used. The subfactors of BSSS are Thrill and Adventure Seeking, Experience Seeking, Disinhibition, and Boredom Susceptibility. The BSSS consists of 20 questions, and responses are recorded on a five-point Likert scale. The higher the total score on the scale, the stronger the sensation seeking tendency. The Cronbach’s alpha value in the previous study was 0.85 [25], and in this study, it was 0.86.

#### 2.3.2. Life Satisfaction Expectancy

The Life Satisfaction Expectancy Scale (LSES) developed by Kim (2007) was used [27]. It consists of 5 questions answered on a 7-point Likert scale. Higher scores indicate higher positive expectations for life. The Cronbach’s alpha value in the previous study was 0.89 [27], and in this study, it was 0.87.

#### 2.3.3. Distress Tolerance

The Korean version [28] of the Distress Intolerance Index (DII) developed by McHug and Otto (2012) [29] was used. It consists of 10 questions, and responses are recorded on a 5-point Likert scale. A higher distress tolerance score indicates a stronger ability to withstand psychological pain. The Cronbach’s alpha value in the previous study was 0.87 [28], and in this study, it was 0.85.

#### 2.3.4. Stock Addiction Tendency

The Stock Addiction Inventory (SAI) developed by Youn et al. [24] was used. The sub-factors of SAI were 6 items for features of problem gambling and 3 items for core features of an addictive disorder, for a total of 9 items. The scale was measured using a 5-point Likert scale. The higher the score, the stronger the tendency to engage in gambling and speculative stock investment. The Cronbach’s alpha value in the previous study was 0.89 [24], and in this study, it was 0.88.

### 2.4. Data Analysis

The collected data were analyzed using IBM SPSS Statistics for Windows (version 25.0; IBM Corp., Armonk, NY, USA). Descriptive statistics of the variables were calculated, and Cronbach’s α was used to assess the reliability of the scale. Pearson’s product correlation analysis was performed for the relationship between the main study variables, and hierarchical regression analysis was used to test the moderating effect.

## 3. Results

### 3.1. Descriptive Statistics of Research Variables

We calculated the mean, standard deviation, skewness, and kurtosis values for sensation seeking, life satisfaction expectancy, distress tolerance, and stock addiction tendency (Table 1). The absolute values of skewness and kurtosis of the study variables were all less than 1.03, assuming normality of the research variables.

### 3.2. Stock Investment-Related Activities

Table 2 presents the results of the correlation analysis of the main study variables. Sensation seeking tendency was significantly negatively correlated with life satisfaction expectancy (*r* = −0.18, *p* < 0.01) and distress tolerance (*r* = −0.21, *p* < 0.01) and significantly positively correlated with stock addiction tendency (*r* = 0.53, *p* < 0.001). Life satisfaction expectancy showed a significant positive correlation with distress tolerance (*r* = 0.31, *p* < 0.001) and a significant negative correlation with stock addiction tendency (*r* = −0.38, *p* < 0.001). Distress tolerance was significantly negatively correlated with stock addiction tendency (*r* = −0.45, *p* < 0.001).

### 3.3. Result of the Moderating Effect of Distress Tolerance on Stock Addiction Tendency

Table 3 shows the moderating effect of distress tolerance on the effects of adults’ sensation seeking tendency and life satisfaction expectancy on stock addiction tendency. The moderating effect was analyzed using hierarchical regression analysis. The interaction term was constructed by multiplying the two indicated variables after mean-centering each variable. The Durbin–Watson value of the regression model was 1.67, assuming independence of the error term, and the variance inflation factor (VIF) value was 1.57 or less, so there was no problem of multicollinearity between independent variables. When the explanatory variables were input into Model 1, sensation seeking had a significant positive effect on stock addiction tendency (*B* = 0.45, *p* < 0.001), and life satisfaction expectancy and distress tolerance had a significant negative effect (*B* = −0.14, *p* < 0.001). The explanation for Model 1 was significant at 43.6% (*R*^2^ = 0.436, *F* = 57.47, *p* < 0.001). Therefore, Hypothesis 1, “sensation seeking, life satisfaction expectancy, and distress tolerance will significantly explain stock addiction tendency” was supported.

When the interaction terms were added to Model 2, interaction term 1 of sensation seeking and distress tolerance appeared to be statistically significant (*B* = −0.35, *p* < 0.001), and interaction term 2 of life satisfaction expectancy and distress tolerance was also statistically significant (*B* = 0.10, *p* = 0.021). In other words, distress tolerance had a significant moderating effect on the effect of sensation seeking and life satisfaction expectancy on stock addiction tendencies. The explanation for Model 2 was 50.0%, which was statistically significant (*R*^2^ = 0.500, *F* = 44.13, *p* < 0.001). Therefore, Hypothesis 2, “there will be a moderating effect of distress tolerance in the effect of sensation seeking on stock addiction tendency”, was supported. Additionally, Hypothesis 3, “there will be a moderating effect of distress tolerance in the effect of life satisfaction expectancy on stock addiction tendency”, was supported.

To clearly understand the moderating effect, the effect of sensation seeking on stock addiction tendency was schematized and presented in Figure 1a according to the ±1 SD level, centering on the average distress tolerance. Looking at Figure 1a, it was confirmed that the stronger the distress tolerance, the weaker the effect of sensation seeking on stock addiction tendency.

We performed Johnson–Neyman significant region analysis to confirm the statistically significant region of the moderating effect. The conditional effect of sensation seeking on stock addiction tendency according to moderating variables and the upper limit (ULCI) and lower limit (LLCI) of the 95% confidence interval is schematized and presented in Figure 1b. Looking at the Johnson–Neyman significant region, it was confirmed that the static effect of sensation seeking on stock addiction tendency was statistically significant in the lower 81.1% of distress tolerance.

When interpreting the results of the Johnson–Neyman significant region analysis, the effect of sensation seeking on stock addiction tendency was not statistically significant in the case of the upper distress tolerance group (19.9%). In other words, a person with strong distress tolerance does not have a stock addiction tendency, even if sensation seeking increases. Therefore, distress tolerance moderated the effect of sensation seeking tendencies on stock addiction tendency.

To clearly understand the moderating effect, the effect of life satisfaction expectancy on stock addiction tendency was schematized and presented in Figure 2a according to the ±1 SD level, centering on the average distress tolerance. Figure 2a confirms that people with low distress tolerance had a stronger stock addiction tendency as their life satisfaction expectancy decreased.

We performed Johnson–Neyman significant region analysis to confirm the statistically significant region of the moderating effect. The conditional effects of life satisfaction expectancy on stock addiction tendency according to moderating variables and the ULCI and LLCI of the 95% confidence interval are schematized and presented in Figure 2b. Looking at the Johnson–Neyman significance region, it was confirmed that the static effect of life satisfaction expectancy on stock addiction tendency was statistically significant in the lower 64.3% region of distress tolerance.

When interpreting the results of the Johnson–Neyman significant region analysis, the effect of life satisfaction expectancy on stock addiction tendency was not statistically significant in the case of the upper distress tolerance group (35.7%). In other words, a person with strong distress tolerance does not develop stock addiction, even if life satisfaction expectancy is lowered. Therefore, distress tolerance moderated the effect of life satisfaction expectancy on stock addiction tendencies.

## 4. Discussion

Sensation seeking had a significant positive effect on stock addiction tendency and distress tolerance had a significant moderating effect. These results were similar to those of previous studies that reported that the higher the sensation seeking disposition, the more likely an individual is to participate in risky behaviors, such as gambling, and to fall into pathological gambling [11,30,31,32]. People with high sensation seeking are impulsive, insensitive to punishment, and sensitive to rewards [33]. A person with high sensation seeking experiences pleasure as an immediate reward according to stock price fluctuations, and this experience motivates them to invest in stocks again. If investors invest in stocks by taking out loans because of lower interest rates, more extreme stress may occur if stock prices are lowered or if interest rates are higher. Impulsive and reward-sensitive people with sensation seeking tendencies react very sensitively even to small changes in stock prices and suffer from difficulties in daily life because they are obsessed with stock thoughts. To solve problems such as gambling, it is necessary to fully understand sensation seeking and exert self-control [34]. It is necessary to establish investment principles such as setting an upper limit on the amount of investment and limiting the amount of time devoted to investing in stocks. Excessive investment in loans entails real problems, so you should check the assets that you can invest in. To make up for losses, you should refrain from continuously increasing investment in risky stocks and come up with alternatives such as receiving expert consulting. Healthy investment behavior should be encouraged in capitalist countries, but it should be properly controlled so that stock investment does not cause impulsive emotions or problems in daily life due to stock investment.

The higher an individual’s negative life expectations, the higher their stock addiction tendency. These results are similar to the results of previous studies showing that life satisfaction expectancy has a significant effect on various behavioral addictions [18,35]. If you experience a sense of economic crisis or have high anxiety about employment, you may become immersed in excessive speculation to improve your negative life. When such negative life expectations arise, stock investment as a countermeasure is not a good choice. Preparing for an uncertain future is something you should undertake in a planned way, and you should not choose impulsive stock investments to avoid anxiety. Frequent fluctuations in stock prices lead investors with low life satisfaction expectancies to experience deeper pain. Therefore, all stock investments require cognitive preparation and practice.

However, stock addiction could be prevented if the distress tolerance to withstand psychological pain was high, even if sensation seeking was high or life satisfaction expectancy was low. If you apply control to impulsive, pleasure-seeking, and stimulating people, they experience frustration and pain. When a stimulating desire is not met, the improvement of distress tolerance that can withstand this is a good alternative for preventing stock addiction. According to previous studies, it has been reported that the poorer the distress tolerance, the more immersed in smartphones one becomes to escape from negative emotional states or the more likely one is to make inappropriate choices such as excessive drinking and smoking [21,36,37,38]. If one cannot endure the psychological pain caused by the negative prospects for the future and the suppression of one’s desires, one can become overly immersed in stock investments that seem to accumulate future wealth while satisfying these stimuli. Therefore, improving distress tolerance by learning adaptive emotional regulation strategies is a good option to prevent stock addiction tendencies.

The limitations of this study and suggestions for follow-up studies are as follows. First, since this study was analyzed based on an online survey, bias in sampling may occur, so care must be taken in interpretation. Second, this study did not conduct a substantial survey of the financial assets of the research participants and could not calculate an accurate value for the investment cost according to income level. Therefore, in a follow-up study, it is suggested to study by controlling for variables according to the individual’s wealth and income level. Third, there was a limitation of the tools used to measure stock addiction. The tools used in this study did not reflect important factors of addiction such as withdrawal symptoms. Therefore, it is proposed to develop a stock addiction measurement tool that reflects various aspects of addiction. In the future, a national investigation agency such as the National Statistical Office can investigate and report stock addiction tendencies, select stock addiction risk groups, and utilize them for prevention and treatment activities. Despite these limitations, this study is meaningful in that it measures the tendency of stock addiction, which has recently increased interest, and reveals some of the psychological causes.

## 5. Conclusions

Even if it involves taking risks, the sensation seeking to pursue stimulating desire and negative life satisfaction expectancy increase stock addiction tendency. However, for those with high distress tolerance, the effects of sensation seeking and negative life satisfaction expectancy on stock addiction tendencies were not significant. Stock addiction tendencies can be prevented by restraining impulsive desires, having positive expectations for one’s life, and enhancing one’s ability to endure psychological pain. In the future, it will be necessary to learn sound stock investment methods and develop a happy life by improving one’s ability to withstand psychological pain.

## Figures and Tables

**Figure 1 behavsci-13-00378-f001:**
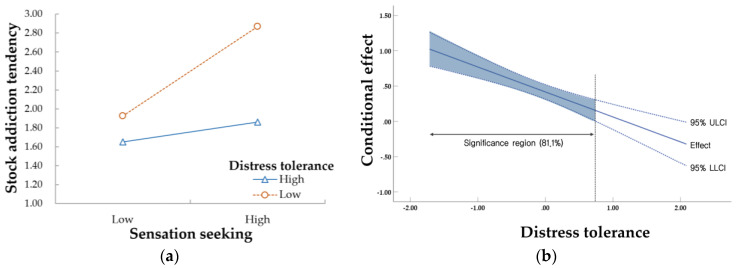
Moderating effect of distress tolerance in the relationship between sensation seeking and stock addiction tendency. (**a**) Conditional effects of distress tolerance level (±1 SD); (**b**) Johnson−Neyman significance region of distress tolerance.

**Figure 2 behavsci-13-00378-f002:**
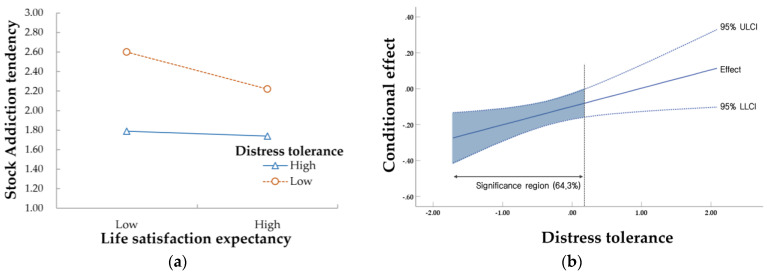
Moderating effect of distress tolerance in the relationship between life satisfaction expectancy and stock addiction tendency. (**a**) Conditional effects of distress tolerance level (±1 SD); (**b**) Johnson–Neyman significance region of distress tolerance.

**Table 1 behavsci-13-00378-t001:** Descriptive statistics of research variables (*N* = 227).

Variables	M ± SD	Skewness	Kurtosis
Sensation seeking	2.81 ± 0.69	−0.12	−0.60
Life satisfaction expectancy	5.26 ± 1.08	−0.41	0.06
Distress tolerance	2.81 ± 0.75	0.38	0.02
Stock addiction tendency	2.11 ± 0.72	0.20	−1.03

**Table 2 behavsci-13-00378-t002:** Correlation coefficient between research variables.

Variables	Sensation Seeking	Life Satisfaction Expectancy	Distress Tolerance
Life satisfaction expectancy	−0.18 **		
Distress tolerance	−0.21 **	0.31 ***	
Stock addiction tendency	0.53 ***	−0.38 ***	−0.45 ***

** *p* < 0.01, *** *p* < 0.001.

**Table 3 behavsci-13-00378-t003:** Results of hierarchical regression analysis on stock addiction tendency (*N* = 227).

Variables	Model 1			Model 2			VIF
	*B*	*SE*	*p*	*B*	*SE*	*p*	
Sensation seeking ^(a)^	0.45	0.06	<0.001	0.42	0.05	<0.001	1.08
Life satisfaction expectancy ^(b)^	−0.14	0.04	<0.001	−0.10	0.04	0.007	1.29
Distress tolerance ^(c)^	−0.28	0.05	<0.001	−0.43	0.06	<0.001	1.57
Interaction 1 * ^(a×c)^				−0.35	0.07	<0.001	1.23
Interaction 2 * ^(b×c)^				0.10	0.04	0.021	1.28
*R* ^2^	0.436 (*F* = 57.47, *p* < 0.001)			0.500 (*F* = 44.13, *p* < 0.001)			

^a,b,c^ Letters indicate each research variable. * The interaction term is the product of mean-centered variables.

## Data Availability

The data presented in this study are not available elsewhere.

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
