# Peer review of "Influence of Sensation Seeking and Life Satisfaction Expectancy on Stock Addiction Tendency: Moderating Effect of Distress Tolerance"

_behavsci, 2023, doi:10.3390/bs13050378_

Round 1

Reviewer 1 Report

This is a timely and important issue to explore that i believ will merit publication. The aims of this study is to test the moderating effect of distress tolerance on the effect of adults’ sensation seeking and life satisfaction expectancy in stock addiction tendency.

The study seems fine, methods are written in details, data are convincing, article language is ok throughout the manuscript. The reasoning and explanationare written satisfactorily. There are very few grammatical mistakes in the manuscript. Methods are written in details which improved the quality of the manuscript

There is, however, issues that must be resolved before the study can be accepted for publication.

- Lack of research hypotheses.

- There is no baseline data as to what the level of stock addiction in this population is at baseline.

- Line 69: „Zuckerman (1994)” - remove the year

- Unfortunately, a link to a Google pole leaves a large amount of bias as to who decides to fill out the survey. Include this in the limitations.

- What kind of a small return gift was provided to the participants?

- Provide inclusion and exclusion criteria.

- Add the strengths of a study and the limitations (limitations include also the not-controlled nature of the study itself and the reliability of responses).

Reviewer 2 Report

Dear Authors,

Thank you for the opportunity to review an interesting article entitled: ‘Influence of Sensation Seeking and Life Satisfaction Expectancy on Stock Addiction Tendency: Moderating Effect of Distress Tolerance’. The aim of this study was to analyse the effect of sensation seeking, corresponding to individual psychological needs, and the expectation of positive life satisfaction on stock addiction tendency, and to verify the moderating effect of distress tolerance.

The strengths of the article presented for evaluation are the statistical analyses used, the citation of current literature.

The manuscript contains only a few errors:

[1].  What is the standard deviation for age? Please add.

[2].  Since the mean age and range is given, a percentage analysis of the different age groups is unnecessary (especially since no analysis of the collected data by age group was performed).

[3].  When describing the characteristics of the subjects, only percentage notation should be used. Giving both the number of individuals and the corresponding percentages reduces the readability of these descriptions (lines 116-131).

[4].  Since the research was carried out via an online survey, it is clear that it was not conducted face-to-face. Hence, highlighting this fact is unnecessary (line 135).

[5].  Please annotate Cronbach's α for both the scales used and in these studies.

[6].  If possible, please add ω McDonalds for your study.

[7].  In line 224, shouldn't it be Figure 1b and not as it is Figure 2b?

[8].  There is no indication of the limitations of the study.
